# Electroporation and Immunotherapy—Unleashing the Abscopal Effect

**DOI:** 10.3390/cancers14122876

**Published:** 2022-06-10

**Authors:** Tobias Freyberg Justesen, Adile Orhan, Hans Raskov, Christian Nolsoe, Ismail Gögenur

**Affiliations:** 1Center for Surgical Science, Zealand University Hospital, Lykkebækvej 1, 4600 Køge, Denmark; aor@regionsjaelland.dk (A.O.); raskov@mail.dk (H.R.); igo@regionsjaelland.dk (I.G.); 2Center for Surgical Ultrasound, Department of Surgery, Zealand University Hospital, Lykkebækvej 1, 4600 Køge, Denmark; cnolsoe@cnolsoe.dk; 3Copenhagen Academy for Medical Education and Simulation (CAMES), University of Copenhagen and the Capital Region of Denmark, Ryesgade 53B, 2100 Copenhagen, Denmark; 4Department of Clinical Medicine, University of Copenhagen, Blegdamsvej 3B, 2200 Copenhagen, Denmark

**Keywords:** electrochemotherapy, irreversible electroporation, immunotherapy, immune response, abscopal effect

## Abstract

**Simple Summary:**

Electrochemotherapy and irreversible electroporation are primarily used for treating patients with cutaneous and subcutaneous tumors and pancreatic cancer, respectively. Increasing numbers of studies have shown that the treatments may elicit an immune response in addition to eliminating the tumor cells. The purpose of this review is to give an in-depth introduction to the electroporation-induced immune response and the local and peripheral immune systems, and to describe the various studies investigating the combination of electroporation and immunotherapy. The review may help guide and inspire the design of future clinical trials investigating the potential synergy of electroporation and immunotherapy in cancer treatment.

**Abstract:**

The discovery of electroporation in 1968 has led to the development of electrochemotherapy (ECT) and irreversible electroporation (IRE). ECT and IRE have been established as treatments of cutaneous and subcutaneous tumors and locally advanced pancreatic cancer, respectively. Interestingly, the treatment modalities have been shown to elicit immunogenic cell death, which in turn can induce an immune response towards the tumor cells. With the dawn of the immunotherapy era, the potential of combining ECT and IRE with immunotherapy has led to the launch of numerous studies. Data from the first clinical trials are promising, and new combination regimes might change the way we treat tumors characterized by low immunogenicity and high levels of immunosuppression, such as melanoma and pancreatic cancer. In this review we will give an introduction to ECT and IRE and discuss the impact on the immune system. Additionally, we will present the results of clinical and preclinical trials, investigating the combination of electroporation modalities and immunotherapy.

## 1. Introduction

In 1968, the first experiments with electroporation were conducted and showed that electric fields increased cell membrane permeability [1]. In 1982, electroporation was demonstrated as an efficient method for transferring DNA into cells [2], thus increasing the uptake of DNA to alter the properties of the cells, which was termed gene electrotransfer (GET). Later in the 1980s, experiments combining electroporation with chemotherapy, i.e., electrochemotherapy (ECT), paved the way for the first clinical trial in 1993 in head and neck squamous cell carcinomas [3]. The use of irreversible electroporation (IRE) was patented in 2003, followed by the first clinical trial in 2010 in prostate cancer [4]. To this day, the major use of electroporation in medicine is within cancer treatment.

### The Abscopal Effect

In 1953, R. H. Mole first coined the term “abscopal”, meaning “away from target”, after documenting the remission of tumors located outside the radiation field in patients with metastatic disease [5]. Since then, almost 50 case reports have been published on the abscopal effect due to radiotherapy alone within various cancers, including melanoma, renal cell carcinoma, and breast cancer [6,7]. The median time to progression was 6 months at the site of the abscopal effect, while potential improvements in survival were not reported. A number of patients showed no recurrence for up to 10 years after achieving abscopal effects with complete response (CR). The abscopal effect is caused by an immune response against tumors that have not been treated directly with, e.g., radiotherapy. However, the rarity of the abscopal effect underlines the difficulties needed to be overcome in order to elicit a significant immune response. Five key events have been linked to effective priming of the T cells: (I) release of tumor-associated antigens (TAAs), (II) release of damage-associated molecular patterns (DAMPs), (III) uptake and processing of TAAs by antigen-presenting cells (APCs), (IV) antigen presentation by APCs to naïve T cells, and finally (V) activation and proliferation of cancer-specific CD8^+^ T cells [7,8]. In recent years, following the introduction of immune checkpoint inhibitors (ICIs) such as anti-programmed cell death receptor/ligand 1 (anti-PD-1/PD-L1) and anti-cytotoxic T lymphocyte antigen 4 (anti-CTLA-4), novel immunotherapy drugs for intratumoral administration have been investigated. These include stimulator of interferon gene (STING) agonists and Toll-like receptor (TLR) agonists, which can be combined with systemic immunostimulatory drugs. To provide an effective immune response and boost the low abscopal effect rates [7], the focus is emerging on combining electroporation modalities with immunotherapy [9,10,11,12]. In this review, we introduce the concepts of ECT and IRE and the interplay between these treatment modalities—the local and systemic immune responses and cancer cells. Finally, we address the current and future perspectives of combining electroporation with immunotherapy.

## 2. Electroporation

Electroporation causes the plasma cell membrane to become permeable to otherwise impermeable molecules due to the exposure of an external electrical current. Electroporation is either reversible or irreversible depending on the duration and the voltage of the electrical pulses (Figure 1). The electric field generated by reversible electroporation temporarily increases the permeability of the cell membrane by the formation of nanopores in the lipid bilayer. This is achieved by applying a series of around eight electrical pulses with a duration of 100 μsec and an amplitude of 100–1000 volts. These parameters depend on a number of factors including the type of electrodes and tumor [12,13,14]. Reversible electroporation in itself only temporarily disrupts cell homeostasis, leaving the cell to fully recover after exposure. This formation of nanopores makes the modality optimal for transferring otherwise insoluble drugs and small molecules into cells [15].

IRE is executed by a large number of pulses ranging from 80 to 100 and up to 3000 volts [16]. Due to the longer duration of exposure and higher amplitude, the treated cells cannot regain homeostasis and undergo cell death. The cell death is non-selective and occurs through both apoptosis and necrosis by irreversible disruption of the membrane and the transfer of ATP and electrolytes [17].

The type of cell death initiated by electroporation is important for its ability to elicit immunogenic cell death (ICD) [18]. ICD is associated with the release of DAMPs, most importantly ATP, high mobility group box 1 (HMGB1) protein, and calreticulin [19]. ATP acts as a “homing” signal, attracting and activating dendritic cells (DCs) [20]; HMGB1 binds to TLR4 and enhances the processing and presentation of TAAs by DCs [21]; and calreticulin acts as an “eat me” signal for phagocytes [22]. Apoptosis is generally considered a non-immunogenic form of cell death; however, caspases can up- or downregulate the release of DAMPs [23], leading to immunogenicity [24]. Necrosis causes the release of higher levels of DAMPs than does apoptosis and is considered an immunogenic type of cell death [18]. Surpassing apoptosis and necrosis in eliciting ICD is necroptosis, which is a form of programmed necrosis independent of caspase activity, which more efficiently activates and primes CD8^+^ T cells against tumor cells [25]. Finally, pyroptosis, a highly inflammatory type of programmed cell death as opposed to apoptosis, is associated with a marked release of DAMPs [18,26]. The properties of the way electroporation initiates cell death are also key, as both the amount and preservation of released TAAs impact the generation of tumor antigen-specific CD8^+^ T cells. IRE has been shown to produce a greater release and preservation of proteins compared to thermal therapy (e.g., radiofrequency and microwave), which was correlated to T cell activation and proliferation [8]. Thus, the quality of both ICD and the release of TAAs are linked to the abscopal effect.

In recent years, a technology similar to IRE called nanosecond pulsed electric fields has attracted attention. As the name implies, the range of the impulses is within the nanosecond range, which might potentially improve the pulse energy control and reduce the muscle contractions related to the IRE treatment [27]. However, due to limited use of the technique and sparse literature available, this will not be further discussed in this review [28]. Finally, it has been proposed to use adjuvant calcium and thereby either lower the electric field thresholds of IRE to reduce potential thermal damage or to induce cell death of the reversibly electroporated cells in the periphery of the tumors [29].

### 2.1. Electrochemotherapy

Bleomycin and cisplatin are the most used chemotherapy drugs in combination with reversible electroporation. Both drugs possess two central properties; they both diffuse poorly across the cell membrane and are highly cytotoxic inside cells, making them optimal for ECT. Bleomycin and cisplatin mainly exert their cytotoxicity by damaging nuclear DNA via forming double-stranded DNA crosslinks and inducing DNA strand breaks, respectively [30]. Administering bleomycin with reversible electroporation decreases the inhibitory concentration several hundred fold [31]. It potentiates the cytotoxicity of bleomycin up to 5000 fold and that of cisplatin up to 12 fold [13]. Additionally, due to the low dosage of the drugs, fewer adverse reactions are seen [32].

In Europe alone, ECT as a monotherapy is being used in 140 cancer centers treating numerous types of cancers, including liver and pancreatic tumors [33,34,35]; melanoma, Kaposi sarcoma; and breast, renal cell, and basal cell carcinoma [36], though it is mainly being used for treating cutaneous and subcutaneous tumors in a palliative setting [37,38]. The procedure has been introduced into European clinical guidelines in melanoma for inoperable skin metastases or primary tumors of the limbs [39,40] and in primary squamous cell carcinoma for inoperable, locally advanced lesions [41]. Further, the Italian Society of Orthopedics and Traumatology has included the procedure in guidelines for unresectable bone metastases of the sacrum [42].

### 2.2. Irreversible Electroporation

IRE is primarily used for treating solid tumors that are unresectable and where thermal ablation or radiotherapy are contra-indicated due to the close proximity of vital structures such as large blood vessels. Due to its mainly non-thermal properties, IRE preserves structures such as blood vessels and biliary tracts by sparing connective tissue and the integrity of the surrounding healthy tissue. The safety and efficacy of IRE have been investigated in several cancers [16], with the most promising overall survival (OS) and recurrence results in pancreatic [43,44], liver [45], and prostate cancer [46,47]. In particular, locally advanced pancreatic cancer represents the most immediate and biggest perspectives of IRE [48]. However, despite the numerous phase I and II trials, larger prospective registries and randomized controlled trials (RCT) directly comparing IRE with standard of care treatment are warranted before IRE can become a widely used cancer treatment modality.

## 3. Modulation of the Immune System

### 3.1. The Interplay between Cancer and Immune Cells

The immune system’s recognition and elimination of malignant cells is of major importance in the development and progression of cancer [49], and its key role in cancer biology and therapy is becoming increasingly more evident [50,51]. Immune evasion, a well-described hallmark of cancer, has gained great research interest during the past decade, and the increasing use of immunotherapy for certain types of solid cancers, such as non-small cell lung cancer, melanoma, breast cancer, and colorectal cancer [52], has made this even more evident. However, despite the development of novel immunotherapy modalities, such as ICIs and adaptive cell transfer for different types of cancer, as well as the continuous research in the field of immuno-oncology, most solid cancers are less responsive or resistant to immunotherapy [53,54]. In continuation, patients with the same type of cancer may respond dissimilarly to identical therapies [54]. The response variations to immunotherapy across cancers and between patients with the same cancer emphasizes the complexity of tumor biology. The constitution of the local tumor microenvironment (TME) as well as the responsiveness of peripheral immune cells may shift the balance towards either immune elimination or immune evasion of cancer cells [55]. This mechanism is very complex and involves the changing interplay between cancer cells, immune cells, and the physical properties of the TME. Here, we will describe immune responses related to cancer cells located in the primary tumor site and in the periphery (Figure 2).

### 3.2. The Local Immune Response to Cancer

When immune cells, especially lymphocytes but also natural killer cells (NK cells), recognize the presence of malignant cells, they can infiltrate the tumor site to eliminate cancer cells through various killing mechanisms. The immune cells that are primarily involved in this process are the CD8^+^ T cells and the CD4^+^ T cells. Both cell types are central components of the adaptive immune system and are activated by APCs of the innate immune system. On the other hand, NK cells of the innate immune system do not depend on activation by APCs to elicit cancer-killing capabilities through cell lysis and the secretion of cytokines [56]. The activation of CD8^+^ and CD4^+^ T cells is highly dependent on signals from the APCs, for which the direct cell-to-cell contact is essential. Activated APCs take up antigens or epitopes and present them to T cells, thereby activating adaptive responses and T cell infiltration. Thus, cancer immunosurveillance is highly dependent on both adaptive and innate immune responses and the interplay between them [57].

Abnormal innate and adaptive immune responses contribute to the development of malignant tumors. Failed immune responses aid in the selection of cancer cell clones that are less likely to undergo immune destruction [58]. Selection pressure and a tumor promoting TME ensure continuous growth and survival advantage. The TME consists of various cell types and signaling molecules that influence cancer progression as well as responses to anti-neoplastic treatments. High levels of CD8^+^ T cells at the tumor site have been correlated with improved survival outcomes in various cancers, including colorectal, breast, and pancreatic cancer [59,60,61]. On the other hand, high levels of the immunosuppressive FoxP3^+^ regulatory T cells (Tregs) and myeloid-derived suppressor cells (MDSC) at the tumor site have been associated with worse prognosis in cancer [62]. Normally, Tregs ensure immune tolerance towards the body’s own cells and prevent the development of autoimmune diseases. The immune responses are continuously calibrated through the secretion of pro- and anti-inflammatory mediators such as interleukins and chemokines. Ultimately, a balance is maintained between the elimination of abnormal cells and the survival of healthy cells under normal circumstances. Through tumor secreted factors and tumor secreted exosomes, cancer cells manage to tip this balance towards a higher infiltration of Tregs and thereby towards an immunosuppressive TME. Together with other immune cell subsets, such as the tumor-associated macrophages (TAMs), the Tregs can compose an immune-evasive TME to enhance the survival and proliferation of cancer cells. Tumor secreted factors and tumor secreted exosomes factors cause an expansion of MDSC, known to possess powerful immunosuppressive properties and are attracted to the TME by chemokine gradients. Tregs, MDSC, and TAMs are known to secrete interleukin 10 (IL-10), transforming growth-factor β (TGF-β), vascular endothelial growth factor, and prostaglandins, leading to immune evasion, neo-angiogenesis, cancer cell proliferation, migration, and survival [63,64]. Chemokines such as CC chemokine ligand 2 (CCL2), CCL5, and vascular endothelial growth factor can recruit TAMs to the tumor-site, thus contributing to the maintenance of an immunosuppressive TME [65]. Likewise, CCL5 can attract Tregs to the tumor-site, whereas high levels of IL-10 and TGF-β in the TME stimulate the differentiation of naïve T cells into Tregs. Thus, the anti-inflammatory signals from the TME can reinforce the Tregs and TAMs, whereas the Tregs and TAMs simultaneously increase the levels of anti-inflammatory signals, resulting in a chain reaction of immunosuppression [66]. Furthermore, TGF-β can activate fibroblasts, especially cancer-associated fibroblasts in the TME, thereby increasing the production of collagen and ultimately desmoplasia. The desmoplastic reaction is the dense extracellular matrix within the tumor that creates a physical barrier for infiltrating immune cells. The activity and efficacy of cytotoxic immune cells are reliant on the direct contact between cancer cells and the lymphocytes [67]. Without this direct cell-to-cell contact, the CD8^+^ T cells have no chance of eliminating cancer cells, and the response to immunotherapy such as ICIs is considerably reduced or absent [54]. However, even when T cells are able to infiltrate the tumor site, their survival and ability to expand can be a serious challenge. The TME is often dominated by hypoxia, acidity, and low nutrient levels, which impair the expansion and survival of immune cells. Thus, the various components of the TME and the interplay between immune cells and stromal cells highly affect the resilience of cancer cells.

### 3.3. The Peripheral Immune Response to Cancer

DAMPs can be released from stressed, damaged, or dying cells and are recognized among others by APCs such as DCs, macrophages, and neutrophils through pattern-recognition receptors, including TLRs. This recognition typically occurs in the peripheral tissue. Once APCs have been activated by innate immune responses such as DAMPs, they can migrate to secondary lymphoid tissue to present and activate lymphocytes against the danger perceived. Thus, the binding of DAMPs to pattern recognition receptors initiates an inflammatory response that ultimately can activate adaptive immune responses involving the T and B lymphocytes [68].

In cancer, DAMPs may be released in response to high cell turnover, cellular stress, or anti-neoplastic treatments. Released DAMPs from tumor cells can bind to the pattern-recognition receptors on the APCs, thereby stimulating the APCs to present endocytosed antigens to CD8^+^ and CD4^+^ T cells through major histocompatibility complexes (MHCs). Once presented with an antigen and activated by APCs, the T cells may proliferate and release further cytokines. By following chemokine gradients and homing receptors, e.g., CD103 for the intestines and cutaneous lymphocyte antigens for the skin, they migrate towards not only the target site but also potential metastatic sites.

Important pro-inflammatory cytokines are IL-2 and IL-15. IL-2 promotes the expansion and activation of NK cells, CD4^+^, and CD8^+^ T cells, while IL-15 activates DCs, stimulates the proliferation of T cells, and enhances the development of NK cells, cells which are all involved in the cancer immuno-surveillance [69]. Activated NK cells and T cells can also release interferon γ (IFN-γ), which is important for the activation of macrophages. IFN-γ also upregulates the expression of MHC-II molecules that are central for the activation of specialized adaptive immune responses.

In summary, the activation of the adaptive immune system against cancer cells, through APCs of the innate immune system, is highly affected by the presence of DAMPs. Once the adaptive immune system has been warned against cancer cells, activated T cells can induce further specialized immune responses and recruit more immune cells through the release of pro-inflammatory cytokines. However, in cancer, the balance between pro- and anti-inflammatory signals is tipped towards anti-inflammation and immunosuppression, mainly due to the properties of stromal cells in the TME. Tumor-secreted factors and tumor-secreted exosomes affect the systemic immune responses to cancer, whereas the systemic immune responses may change the constitution of the TME. Ideally, multimodal treatment approaches targeting both the local and systemic changes related to tumorigenesis and dissemination could be promising.

### 3.4. ECT and the Immune System

Preclinical studies of immunocompetent versus immunodeficient mice have established that the efficacy of ECT depends on the competency of the immune system [70,71]. ECT induces ICD through the liberation of ATP, HMGB1, and calreticulin [71], which in turn might increase the tumor infiltration of several immune cells, including CD8^+^ T cells and NK cells [72,73,74]. The number of tumor-infiltrating NK cells increased up to four fold following ECT in both preclinical and clinical studies [72,74,75]. In addition, ECT has been shown to preserve large blood vessels [76,77] and to enhance the release of TAAs [74]. ECT can induce an anti-tumor immune response and has been shown to repress distant tumor growth of a non-treated lesion in a murine model of colorectal cancer [75] and to induce systemic responses in murine models [73], thus eliciting abscopal effects, findings that have not yet been replicated in clinical studies [78,79] (Table 1).

### 3.5. IRE and its Effects on the Immune System

IRE induces an ICD by the release of DAMPs from tumor cell apoptosis [85,86,87,88]; however, more inflammatory types of cell death, such as necrosis, necroptosis, and pyroptosis, are also linked to ICD [89]. These different findings may in part be explained by differences in the cell-death induction between cell types [18] and the presence of more than one type of cell death [90,91], while it may also be due to differences or limitations in the way cell death is assessed [92]. Nonetheless, several properties of IRE do increase the likelihood of activating the immune system against cancer. First, preservation of the larger vessels and increased microvessel density allow the APCs to infiltrate the treated lesion, carry tumor antigens to draining lymph nodes, and activate the adaptive immune system and the subsequent immune infiltration [93,94,95]. Second, the release of large quantities of tumor-associated antigens can be taken up by APCs. These antigens are highly preserved compared to heat-based therapies (radiofrequency and microwave) [8]. Third, the release of DAMPs including ATP, HMGB1, HSP70, and calreticulin from tumor cells are vital for inducing ICD [87,88]. In addition, HMGB1 released from IRE-treated tumor cells have been shown to reprogram TAMs from immune-suppressive (M2) to immune-promoting (M1) phenotypes [88], thereby increasing the M1/M2 ratio in the TME and the periphery [96]. Fourth, immune suppression may be counteracted through lower-level Tregs and MDSCs, both peripherally and in the TME [96,97]. Finally, modulation of the tumor stroma may increase the infiltration of immune cells by increasing the microvessel density and decreasing the rigidity of the extracellular matrix [95]. These properties, as well as the abscopal effects [98], can be exploited in the preclinical setting of micrometastatic or metastatic disease, as studies indicate a significant immune response, including enhanced immune memory [95,99]. Investigations of immunocompetent and immunodeficient mice have shown that a responsive immune system is vital for the optimal efficacy of IRE [100], although a murine sarcoma model did not reveal any tumor-infiltrating CD4^+^ or CD8^+^ T lymphocytes within six hours after IRE [101]. IRE-treated tumors have shown increased infiltration of CD30-positive cells [98], which indicate the presence of lymphocytes after primary allo-antigenic stimulation [102] and undifferentiated pluripotent stem cells [103]. This indicate an enhanced immune response via TAA activation of T cells or recruitment of stem cells by residual tumor cells. However, until further studies elucidate these mechanisms, the impact of this finding remains unknown. The key findings of the effects of IRE on the immune system are summarized in Table 2.

The cytokine levels of IL-2, IL-6, and IL-10 have been shown to be elevated up to 7 days after IRE [94,104], while IL-2 and IL-10 have been shown to decrease 3–21 days after IRE [105]. IL-2 is associated with tumor inhibiting properties by the modulation of lymphocyte proliferation and function [106,107]. IL-6 is associated with increased angiogenesis and subsequent tumorigeneses via different pathways, in particular the Janus kinase-signal transducer and activator of transcription 3 pathway [108]. Finally, IL-10 may play an immunosuppressive role in the TME via increased expression of B7-H4 on macrophages and PD-L1 on monocytes [109,110,111]. It is not clear if the combined effect of the cytokine levels are tumor promoting or not. This may depend on the dose relationship, time-dependent changes in levels, and the interplay with other cytokines and transcription factors [109,111].

**Table 2 cancers-14-02876-t002:** Summary of IRE studies investigating the effects on the immune system.

Species	Authors	Interventions (n)	Cancer Types	Key Findings
Human	Guo et al., 2021 [112]	IRE (11)	Hepatocellular carcinoma	The peripheral neutrophils and monocytes increased by day 1 after IRE and returned to baseline at day 7, while CD4^+^ T cells decreased by day 1 followed by an increase in the next days. CD8^+^ T cells remained unchanged. Treg cells decreased from day 3 to 14 followed by an increase at one month.
	He et al., 2019 [104]	IRE (34)	Locally advanced pancreatic cancer	The peripheral CD4^+^ T cells, CD8^+^ T cells, and NK cells decreased by day 3 after IRE followed by an increase at day 7, while a reverse trend was shown for Treg cells. IL-6 and IL-10 levels increased at day 3 after IRE followed by a decrease at day 7. IL-2 increased from day 3 to day 7. Concentrations of IFN-γ and TNF did not significantly change. Increased numbers of CD4^+^ T cells, CD8^+^ T cells, and NK cells or decreased Treg cells were associated with longer OS.
	Pandit et al., 2019 [97]	IRE/pancreatectomy (11/4)	Locally advanced pancreatic cancer	The peripheral Treg populations increased day 1 to 3 and decreased from day 3 to 5 in the IRE group compared to increases on day 1 to 3 as well as increases on day 3 to 5 in the pancreatectomy group.
	Scheffer et al., 2019 [113]	IRE (10)	Locally advanced pancreatic cancer	Pre- and post-IRE peripheral levels of CD4^+^ and CD8^+^ T cells did not change. At 2 weeks following IRE, a decrease in total Tregs was observed, as well as in aTregs and in resting Tregs, accompanied by a transient increase in both peripheral CD4^+^PD-1^+^ and CD8^+^PD-1^+^ T cell numbers.
	Beitel-White et al., 2019 [114]	IRE (8)	Pancreatic cancer (stage III)	An increase in current change during IRE treatment was associated with decreases in Treg populations 24 h after IRE. Changes in current above 20A induced decreased Treg populations. Further, a trend was shown towards increased survival for the group of patients with a >2% decrease in Treg cells.
Swine	Fujimori et al., 2021 [115]	IRE/microwave ablation	Normal lung	Fifty percent of blood vessels and collagen were intact 2 days after IRE compared to 0% after microwave ablation. Further, the number of CD3^+^ T cells increased more after IRE than after microwave ablation.
Rabbit	Lee et al., 2012 [98]	IRE	Hepatocellular carcinoma	Examinations of non-IRE treated organs, e.g., the lungs, showed no metastases in the IRE group, while all 15 rabbits in the control group had lung metastases. IRE-treated tumors showed increased levels of CD30-positive cells, mainly in the zone between viable and dead tumor.
Mouse	Dai et al., 2021 [99]	IRE	Hepatocellular carcinoma	IRE increased the percentage of IFN-γ^+^ CD8^+^ T cells in splenocytes and increased tumor infiltration of CD8^+^ T cells. On day 7, reductions of both peripheral and intratumoral Treg cells and PD-1^+^ T cells were shown. Mice rejected the tumor re-challenge with hepatocellular carcinoma cells following IRE.
	He et al., 2020 [95]	IRE	Pancreatic cancer	IRE resulted in longer survival and more proliferating CD8^+^ T cells in the tumor and spleen. Both memory and effector CD8^+^ T cells were increased in the tumor and the tumor-draining lymph node regions. The viable region showed increased microvessel density and softening of the extracellular matrix. Mice that were re-challenged with pancreatic cancer cells after IRE rejected the tumor challenge.
	Chen et al., 2017 [116]	IRE	Hepatocellular carcinoma	IRE induced a change in the T helper 1/T helper 2 cell ratio towards T helper 1 dominance, an increase in macrophage tumor infiltration, and an increase in IFN-γ and IL-2 compared to controls.
	White et al., 2018 [117]	IRE or cryoablation	Pancreatic cancer	IRE induced a higher number of tumor-infiltrating T cells and macrophages at 12 and 24 h after treatment.
	Bulvik et al., 2016 [94]	IRE/radiofrequency ablation (82/82)	Normal liver Hepatocellular carcinoma	The tumor infiltration of neutrophils and macrophages was increased in both groups; however, it was greater in the radiofrequency ablation group. In the IRE group, the infiltration of the neutrophils and macrophages extended along the preserved vessels within the ablation zone. At 72 h, persistent vessels in the ablation zone were seen for IRE-treated mice only. IL-6 levels peaked after 6 h, 3 and 10 times higher than controls (radiofrequency ablation and IRE, respectively). By 24 h, no elevations were seen. Radiofrequency ablation of the liver slowed the growth of an untreated tumor, while IRE resulted in greater reduction in tumor growth. Three days after treatment, the number of CD3^+^ cells was elevated in the untreated tumor in both groups.
	Neal et al., 2013 [100]	IRE	Renal carcinoma	IRE-treated immunocompetent mice showed robust T-cell infiltration at the zone between viable and dead tumors. Further, IRE-treated immunocompetent mice showed a greater treatment response than did immunodeficient mice.
	José et al., 2012 [93]	IRE	Pancreatic cancer	IRE was not found to activate apoptotic cell death measured by caspase-3 positive cells in the tumors. The vascular architecture of the tumor was disrupted from day 1 after IRE and onward.
	Al-sakere et al., 2007 [101]	IRE	Sarcoma	No tumor infiltration of CD4^+^ or CD8^+^ T lymphocytes, macrophages, APCs, dendritic cells were observed 2 and 6 h after IRE.
	Li et al., 2012 [105]	IRE/sham surgery/resection/control (28/28/28/28)	Osteosarcoma	IRE and resection increased the percentages of the peripheral CD3^+^ and CD4^+^ cells, as well as the CD4^+^/CD8^+^ ratio 7 days after treatment. A more rapid and prolonged increase was seen in the IRE group. IRE and resection caused decreases in IL-10 from day 3 to 21. The percentage of INF-γ-positive splenocytes was higher in the IRE group.
Rat	He et al., 2021 [88]	IRE	Pancreatic cancer	IRE caused increased levels of HMGB1, HSP70, and calreticulin. Seven days after IRE, higher frequencies of M1 macrophages in the tumor and a regional lymph node were seen compared to controls, while a decrease in M2 macrophages was seen in the tumor.
Cell	He et al., 2021 [88]	IRE	Pancreatic cancer	HMGB1 were shown to induce M1 macrophage polarization via receptor of advanced glycation end-product. Further, HMGB1 could enhance the phagocytosis of dying tumor cells by macrophages.
	Shao et al., 2019 [8]	IRE/thermal therapy/cryosurgery	Melanoma	IRE caused the greatest protein release, second lowest denaturation rate of the released protein (30%), the most TLR2 (a measure of the relative antigen content of the released protein) release, and the strongest T cell response.
	Zhao et al., 2019 [87]	IRE/radiotherapy	Pancreatic cancer Melanoma	IRE increased the ATP and HMGB1 levels by 11 and 13 fold, respectively, compared to radiotherapy, which did not cause the release of ATP and HMGB1. IRE: Cells increased the expression of makers for DC activation/maturation by 51–72%, compared to non-IRE treated cells. Radiotherapy: Cells did not increase the expression of makers for DC activation/maturation, compared to non-radiotherapy treated cells. IRE increased the ATP and HMGB1 levels by 8 and 9 fold, respectively.
	Goswami et al., 2017 [118]	IRE/thermal shock/chemical poration	Triple negative breast cancer	IRE caused upregulation of IL-6 and TNF, while thymic stromal lymphopoietin was down-regulated. Cancer cells treated with thermal shock or chemical poration showed no down-regulation of thymic stromal lymphopoietin.

APC, antigen-presenting cell; aTreg, activated Tregs; DC, dendritic cell; IL, interleukin; IRE, irreversible electroporation; Treg, regulatory T cell; TRP2, Toll-like receptor 2.

## 4. The Synergy of Electroporation and Immunotherapy

### 4.1. Immunotherapy

Immunotherapy is the broad term for therapies that augment anti-cancer immune responses. The most prominent immunotherapies include ICIs, cytokines, and cell transfers. ICIs target inhibitory receptors located on T cells, such as CTLA-4 and PD-1, as well as their ligands on tumors cells, e.g., anti-PD-L1 and anti-CTLA-4. ICIs prevent T cell exhaustion and apoptosis, thereby enhancing cytotoxicity [52]. ICIs have changed the cancer treatment landscape in a number of cancer types since the first drug was approved by the FDA in 2011 [119]. However, the remarkable efficacy seen in some cancers has been limited to subgroups of tumors. They have been characterized by high expression of PD-1/PD-L1, a high mutational burden, or by a deficient mismatch repair system, together commonly known as immunogenically “hot” tumors. “Cold” tumors on the contrary respond poorly to ICIs and are characterized by being sparsely infiltrated or devoid of immune cells [52].

The most prominent cytokine in immunotherapy is IL-2, a T cell proliferation factor which may also enhance the cytotoxicity of NK cells [106]. Today, IL-2 is no longer the most widely used immunotherapy drug, in part due to the potential life-threatening adverse reactions when used in high doses [120] as well as the approval of novel immunotherapies including ICIs. NK cell transfer therapy (NK cell transfer) is one of the cell transfer therapies that has been known for the longest. It utilizes the inherent tumor killing capabilities of the NK cells, which are independent of MHC molecule presentation [121]. NK cells do not require HLA matching, and promising early results have been reported using allogenic NK cell transfer for the treatment of acute myeloid leukemia [120]; however, indications are limited. Today there are several FDA-approved T cell transfer therapies. However, chimeric antigen receptor T cell therapies are limited due to manufacturing processes and cost and treatment-related toxicity.

### 4.2. ECT and the Synergy with Immunotherapy

In 2003, the first study investigating ECT in combination with immunotherapy was published [122]. Since then, several small-scale studies investigating different types of immunotherapy in combination with ECT have been conducted. The clinical trials have been focused on patients with advanced or metastatic melanoma, combining ECT with either anti-PD-1, anti-CTLA-4, INF-α, or low-dose IL-2. Figure 3 shows how ECT or IRE in combination with immunotherapy may induce a local immune response and an abscopal effect to increase the local and systemic tumor responses. In 2016, the largest prospective study in patients with advanced malignant melanoma was published [123]. It found significantly increased OS among the patients that received both local therapy (ECT, radiotherapy, or stereotactic radiation) + ipilimumab (anti-CTLA-4) compared to ipilimumab alone (median OS of 93 weeks vs. 42 weeks), though it should be noted that only 4 in 45 patients received ECT treatment. Recently, the largest retrospective study to date included 130 patients with metastatic melanoma [124]. Data from the European InspECT group was combined with Slovenian registry data. Local tumor response rates were significantly higher in the group of patients that received either ECT or ECT + pembrolizumab (anti-PD-1) compared to pembrolizumab alone. Systemic tumor response rates were comparable between ECT and ECT + pembrolizumab; however, the time to progression was doubled among the patients that received ECT + pembrolizumab (8 and 17 months, respectively, *p* < 0.05). Survival data showed 2-year overall survival (OS) of 43% and 70% in the ECT and ECT + pembrolizumab groups, respectively. The chance of survival was significantly higher in the ECT + pembrolizumab group, when data were adjusted for previous systemic therapies. No serious adverse events were reported in the treatment groups. A retrospective study by Heppt et al. [125] of 33 patients with metastatic melanoma showed 15 months median OS among the five patients that received ECT + anti-PD-1, while the median OS in the ECT + ipilimumab group was not reached. The local and systemic objective response rates were comparable to those in the study by Campana et al. [124].

IFN-α administered as a post-surgical adjuvant before ECT treatment was investigated in a small retrospective study in five patients with recurrent melanoma [126]. Two patients received low-dose IFN-α, while three patients received high-dose IFN-α. The time interval between the end of IFN-α treatment and ECT treatment ranged from 7 months to 12 years, and all patients presented with metastatic disease at the time of ECT treatment. Three patients with 1–23 metastatic lesions achieved CR 4 weeks after ECT, one patient achieved CR in >85% of 80 lesions, and one patient achieved 100% PR in five lesions. No adverse events were reported. These findings indicate that IFN-α and subsequent ECT could be a safe and effective treatment combination, though the study had many limitations, including a retrospective design and a small heterogeneous group of patients both regarding IFN-α dose and treatment interval.

In preclinical studies, ECT has been investigated in combination with IL-12 GET, showing improved CR rates compared to ECT alone [81,127,128,129]. Interestingly, a multi-arm study found the added efficacy of IL-12 to be the greatest in the poorly immunogenic melanoma model (0% CR vs. 38%, alone and combined treatment, respectively) [127]. The immune status pre-treatment was evaluated by CD4^+^ and CD8^+^ T cell tumor infiltration and expression of MHC-1 and PD-L1. Promising preclinical results of ECT in combination with either inducible T-cell co-stimulator activating antibody [130], TNF-α [131,132], or CpG oligodeoxynucleotides [73] have yet to enter clinical trials, so conclusions cannot yet be drawn. Antibodies targeting the inducible T-cell co-stimulator receptor located on T cells are novel and result in the activation and expansion of anti-tumor T cells. In summary, clinical trials of ECT and ICI have shown promising results with systemic responses indicating abscopal effects in melanoma patients, although larger RCTs are warranted (Table 3).

### 4.3. IRE Plus Immunotherapy

In recent years, several clinical studies have investigated IRE in combination with either ICIs or immune cell transfer therapy (Table 4). These studies have primarily been conducted on patients with locally advanced pancreatic cancer, likely due to IRE as monotherapy already being indicated in this patient population. One prospective trial of 10 patients with locally advanced pancreatic cancer treated with IRE + anti-PD-1 showed a median OS of 18 months [140]. This contrasts with a retrospective study of IRE + anti-PD-1, which showed a median OS of 44 months in locally advanced pancreatic cancer [141]. Small sample sizes and different study designs make the studies difficult to compare; however, the retrospective study did show a superior OS in the IRE + anti-PD-1 group versus IRE alone (median OS 44 months vs. 23 months, respectively). In addition, CD4^+^ and CD8^+^ T cell numbers increased in the IRE + anti-PD-1 group. Finally, the levels of IL-4, IL-6, TNF, and IFN-γ increased more in the combination group than in the IRE monotherapy group [141], indicating a potential survival and molecular benefit when combining IRE with anti-PD-1. The combination of IRE and allogenic NK cell transfer in locally advanced pancreatic cancer showed significantly improved response rates compared to IRE alone in an RCT [142]. However, the response rates did not translate into significantly improved survival. Higher numbers of CD4^+^ and CD8^+^ T cells, NK cells, and B cells were also seen in the IRE + NK cell transfer group compared to IRE alone. A prospective trial showed significantly improved survival data in the same patient population of IRE + NK cell transfer compared to IRE alone (median OS 13.2 months and 11.4 months, respectively) [143]. Thus, evidence indicates added efficacy when transferring NK cells in combination with the IRE treatment, though the gain may be minor. An RCT investigating the use of IRE and allogenic Vγ9Vδ2 T cell transfer showed more promising results in locally advanced pancreatic cancer [144]. Significantly increased survival was seen in the IRE + allogenic γδ T cell transfer group versus IRE alone (median OS 14.5 months vs. 11.0 months, respectively). Vγ9Vδ2 T cells are the major subset of peripheral γδ T cells, which share central properties with both T and NK cells. In contrast to αβ T cells (CD4^+^, CD8^+^ T cells), γδ T cells recognize and kill transformed cells by endogenous tumor-derived pyrophosphates. A mechanism independent of the major histocompatibility complex class molecules was displayed by APCs and target cells [145]. In addition, γδ T cells express activating NK receptors, e.g., NKp30, NKp44, and NKG2D, that bind to stress-inducible molecules frequently expressed by cancer cells [146]. Thus, γδ T cells have two independent recognition systems to sense tumor cells and to initiate cytotoxicity and cytokine production. Moreover, patients receiving multiple transfers of γδ T cells had significantly longer median OS compared to patients that only received a single course (17 months vs. 13.5 months, respectively). In addition, trials of adoptive cell transfer therapy in combination with IRE found no significantly added safety issues when adding cell therapy to IRE, which further support continued research in the combinatory regimens [143,144]. γδ T cell therapy does seem superior to NK cells, though the RCTs of the two cell therapies would bring vital information for future studies when deciding the most optimal cell therapy to combine with IRE in locally advanced pancreatic cancer.

The combination of IRE and allogenic NK cell transfer has also been investigated in primary liver cancer (intrahepatic cholangiocarcinoma and hepatocellular carcinoma). An RCT of 40 patients yielded median OS values of 23.2 months and 17.9 months in IRE + NK cell transfer and IRE-treated patients, respectively (*p* < 0.05) [147]. Moreover, a retrospective study of 40 patients with metastatic hepatocellular carcinoma found IRE + NK cell transfer to yield significantly longer survival [148]. Both treatments had benign safety profiles, as no serious adverse events were reported in either group [147,148]. Thus, combining IRE with immunotherapy may have elicited an abscopal effect, resulting in improved survival. 

Preclinical studies have investigated a wide range of immunotherapies administered in various combinations with IRE, including anti-CTLA-4 [149], anti-PD-1 [87,150], TLR agonists [96,151], M1 oncolytic virus [90], and STING agonist [152,153]. Studies have been focused mainly on pancreatic cancer and to a lesser extent hepatocellular carcinoma, melanoma, prostate cancer, and breast cancer. IRE + anti-CTLA-4 and anti-PD-1 have shown CR rates of 46% in prostate cancer along with increased numbers of CD8^+^ T cells specific for the prostate tumor antigen, both locally and peripherally [149]. Further, IRE + anti-PD-L1 + TLR3 and TLR9 agonist have shown CR rates of 100% in both lymphoma and breast cancer models. Moreover, this combination regimen induced tumor infiltration of CD8^+^ and CD4^+^ T cells and increased the ratio of the anti-tumor M1 macrophages, while reducing both the number of Tregs cells and MDSCs [96]. IRE + STING agonist showed limited efficacy in melanoma [152] and great efficacy in hepatocellular carcinoma [152]. Further, in a Lewis lung carcinoma model, IRE + STING increased the tumor infiltration of CD8^+^ and CD4^+^ T cells and induced a shift in the M1/M2 macrophage ratio towards the anti-tumor M1 phenotype [153]. Among the combinations investigated in preclinical studies, the combination of IRE and anti-PD-L1, and the TLR agonist look the most promising, and an explorative study is underway investigating the combination treatment in metastatic pancreatic cancer [154].

**Table 4 cancers-14-02876-t004:** Summary of IRE + immunotherapy studies.

Species	Authors	Interventions (n, Study Design)	Cancer Types	Key Findings
Human	He et al., 2021 [141]	IRE/IRE + toripalimab (70/15) **	Locally advanced pancreatic cancer	Median OS: 1, 2, and 3 year OS rates: IRE: 23.4 months: 91%, 45%, and 12%. IRE + toripalimab: 44.3 months: 100%, 100%, and 33.3%. Increased CD4^+^ and CD8^+^ T cells, while CD8^+^ Treg cells decreased compared to IRE. Further the levels of IL-4, IL-6, TNF, and IFN-γ increased markedly more than in the IRE group.
	Pan et al., 2020 [142]	IRE/IRE + allogenic NK cell transfer (46/46) ****	Locally advanced pancreatic cancer	Median OS: Response rates: IRE: 11.8 months: 15% CR, 41% PR. IRE + NK cells: 12.4 months: 30% CR, 41% PR. Increased CD4^+^, CD8^+^ T cells, NK cells, and B cells compared to IRE alone. Further, the levels of IL-2, TNF-β, and IFN-γ increased markedly more than in the IRE group.
	Lin et al., 2020 [144]	IRE/IRE + allogenic γδ T cell transfer (32/30) ****	Locally advanced pancreatic cancer	Median OS: IRE: 11.0 months. IRE + T cells: 14.5 months. Twenty-five incidences of grade 3/4 adverse events equally distributed in both groups.
	O’Neill et al., 2020 [140]	IRE + nivolumab (10) ***	Locally advanced pancreatic cancer	Median OS: 18 months; 1 year OS: 67%. Adverse events ≥ grade 3: 70% of patients. By day 90, T effector memory cells were increased two fold from baseline.
	Yang et al., 2019 [147]	IRE/IRE + allogenic NK cell transfer (22/18) ****	Unresectable Intrahepatic cholangiocarcinoma/hepatocellular carcinoma	Median OS: Response rates: IRE: 17.9 months: 5% CR, 64% PR IRE + NK cells: 23.2 months: 17% CR, 72% PR Higher lymphocyte count and IL-2, TNF-β, IFN-γ levels post-treatment compared to IRE alone. No serious adverse events.
	Alnaggar et al., 2018 [148]	IRE/IRE + allogenic NK cell transfer (20/20) **	Metastatic hepatocellular carcinoma	Median OS: IRE: 8.9 months. IRE + NK cells: 10.1 months. Lower number of circulating tumor cells in the IRE + NK cell group at 7 and 30 days after treatment. No serious adverse events and no differences in lymphocyte subsets between the two groups after treatment.
	Lin et al., 2017 [143]	IRE/IRE + allogenic NK cell transfer (39/32) ***	Pancreatic cancer (stage III/IV)	Median OS: IRE: 11.4 months (stage III), 8.7 months (stage IV). IRE + NK cells: 13.2 months (stage III), 9.8 months (stage IV). No serious adverse events.
	Lin et al., 2017 [155]	IRE/IRE + allogenic NK cell transfer (19/20) ***	Metastatic pancreatic cancer	IRE: 16% CR, 47% PR IRE + NK cells: 30% CR, 50% PR
Mouse	Burbach et al., 2021 [149]	IRE + anti-CTLA-4 + anti-PD-1	Prostate cancer	IRE/anti-CTLA-4: 0%/15% CR. IRE + anti-CTLA-4: 46% CR Increased number of CD8^+^ T cells both locally and systemically compared to IRE or anti-CTLA-4. IRE + anti-CTLA-4, and subsequent anti-PD-1: Sustained tumor regression after CR.
	Shi et al., 2021 [150]	IRE + anti-PD-L1	Hepatocellular carcinoma	IRE + anti-PD-L1-induced necrosis, T cell and inflammatory cell infiltration in both treated and non-treated tumors.
	Babikr et al., 2021 [96]	IRE + anti-PD-L1 + TLR3 + TLR9 agonists	Lymphoma Breast cancer	IRE: 0% CR. IRE + TLR3 + TLR9: Superior primary tumor growth inhibition and CD8^+^ T cell response compared to IRE and IRE + anti-PD-1. IRE + anti-PD-1 + TLR3 + TLR9: 100% CR of treated and non-treated tumors. Increased the tumor infiltration of CD8^+^ and CD4^+^ T cells and the CD8^+^ T cell response compared to IRE alone. Induced a M1/M2 macrophage balance towards the anti-tumor M1 and reduced Tregs and MDSCs. IRE + anti-PD-1 + TLR3 + TLR9: 100% CR of treated tumors.
	Zhang et al., 2021 [156]	IRE + anti-OX40	Pancreatic cancer Metastatic pancreatic cancer	Median survival: Control/anti-OX40/IRE: 22/24/51 days. IRE + anti-OX40: 80% were alive at 120 days (median survival not reached). Increased tumor infiltration of CD8^+^ T cells and decreased MDSCs, as well as higher levels of IFN-γ and TNF-α compared to IRE alone. Secondary, non-treated tumor, median survival: Control/anti-OX40/IRE: 21/21/31 days, respectively. IRE + anti-OX40: 44 days.
	Sun et al., 2021 [90]	IRE + M1 oncolytic virus	Pancreatic cancer	Median survival: Control/M1 virus/IRE: 31/34/46 days. IRE + M1: 58 days. Increased tumor infiltration of CD4^+^ and CD8^+^ T cells.
	Yang et al., 2021 [157]	IRE + DC vaccine	Pancreatic cancer	Median survival: Control/IRE/DC vaccine: 35/44/49 days. IRE + anti-PD-1: 77 days. Twice as high mean number of tumor-infiltrating CD8^+^ T cells compared to IRE alone.
	Lasarte-Cia et al., 2021 [152]	IRE + STING agonist	Melanoma Hepatocellular carcinoma	Control/IRE/STING: 0% CR. IRE + STING: 13% CR. Control/IRE/STING: 0%/17%/20% CR. IRE + STING: 67% CR.
	Go et al., 2020 [153]	IRE + STING agonist	Lewis lung carcinoma	IRE + STING: Reduced the tumor volume, induced a M1/M2 macrophage balance towards the anti-tumor M1 phenotype, and increased the tumor infiltration of CD8^+^ and CD4^+^ T cells compared to IRE or STING alone.
	Yu et al., 2020 [158]	IRE + indoleamine 2,3-dioxygenase inhibitor loaded electric pulse responsive iron-oxide-nanocube clusters	Prostate cancer	Combination treatment induced higher calreticulin tumor exposure, increased frequency of tumor-infiltrating CD3^+^ T cells, and higher CD8^+^ T cell-to-Tregs ratio compared to IRE alone. Further, it reduced the tumor growth of both treated and non-treated tumors more than IRE alone.
	Narayanan et al., 2019 [151]	IRE + TLR7 agonist/anti-PD-1	Pancreatic cancer	IRE: 20–35% CR in immunocompetent mice; 0% CR in immunodeficient mice. Generated tumor antigen-specific T cell responses. IRE + TLR7/anti-PD-1 were not superior to IRE alone in survival and tumor growth reduction.
	Zhao et al., 2019 [87]	IRE + anti-PD-1 + anti-CTLA-4/radiotherapy + anti-PD-1	Pancreatic cancer Melanoma	Median survival: Control/anti-PD-1/IRE: 6/8/12 days. Radiotherapy + anti-PD-1: 30 days; 0% were alive at 120 days. IRE + anti-PD-1: 32 days; 36% were alive at 120 days. IRE + anti-PD-1 + anti-CTLA-4: 41 days. However, not significantly different from IRE + anti-PD-1, and weight loss suggested considerable toxicity. Median survival: Control/anti-PD-1/IRE: 5/6/8 days. IRE + anti-PD-1: 23 days
	Vivas et al., 2019 [159]	IRE + polyinosinic-polycytidylic acid and poly-L-lysine	Hepatocellular carcinoma	Control/IRE/polyinosinic-polycytidylic acid and poly-L-lysine: 0%/27%/30% CR. IRE + polyinosinic-polycytidylic acid and poly-L-lysine: 71% CR.
	Pasquet et al., 2019 [160]	IRE + IL-12 GET	Melanoma	Control/IRE/IL-12 GET: 0% CR. IRE + IL-12 GET: 42% CR.

** retrospective study; *** prospective study; **** RCT; anti-CTLA-4, cytotoxic T-lymphocyte-associated antigen 4 inhibitor; anti-PD-1, programmed death-1 receptor inhibitor; CR, complete response; DC, dendritic cell; GET, gene electrotransfer; IFN, interferon; IL, interleukin; IRE, irreversible electroporation; MDSC, myeloid-derived suppressor cell; NK, natural killer; OS, overall survival; STING, stimulator of interferon genes; TLR, Toll-like receptor; TNF, tumor necrosis factor; Treg, regulatory T cell.

## 5. Perspectives

### 5.1. Ongoing Trials

One trial is registered on clinicaltrials.gov investigating ECT + pembrolizumab in unresectable melanoma, which aims to include 53 patients (NCT03448666). Several trials investigating IRE in combination with immunotherapy are ongoing. Three trials will include 10 to 18 patients with locally advanced pancreatic cancer or metastatic pancreatic cancer treated with IRE + nivolumab (NCT03080974 and NCT04212026) or pembrolizumab (NCT04835402). One trial will investigate nivolumab, IRE + nivolumab, and IRE + nivolumab + TLR9 agonist in 18 patients with metastatic pancreatic cancer (NCT04612530) [154]. Finally, one trial will investigate IRE + nivolumab in 43 patients with advanced hepatocellular carcinoma (NCT03630640).

### 5.2. Intertumoral Heterogeneity

Increasing numbers of studies are investigating ECT or IRE in combination with immunotherapy to enhance the local immune response and induce an abscopal effect, which depends on the genetic composition of the tumor. Differences in the genetic landscape (intratumoral heterogeneity) caused by constant selection pressure that leads to the survival and growth of the most resilient clones have been known for decades. Thus, intermetastatic heterogeneity, including the composition of the TME, exists among different metastatic lesions in the same patients as well as heterogeneity between the primary tumor and metastases [161,162]. This may be associated with the low concordance of T cell tumor infiltration between primary tumor and matched metastases seen in colorectal cancer [163]. Although typical driver gene mutations are shared by all lesions, a patient with metastatic cancer will have other critical tumor mutations that are not shared [164,165]. Employing multiple-site electroporation can help to ensure (I) generating TAAs specific for all lesions, (II) releasing a sufficient amount of TAAs, and (III) releasing enough DAMPs. This could help overcome the inherent intertumor and intermetastatic heterogeneity and elicit a pronounced immune response [166]. This approach is already commonly used in ECT plus immunotherapy studies, as ECT can readily be used to treat multiple cutaneous and subcutaneous metastatic lesions [125,135]. In contrast, IRE is mainly used to treat single lesions, in part due to numerous studies investing IRE plus immunotherapy in locally advanced pancreatic cancer. Moving forward, we propose designing studies that aim at treating two or more lesions simultaneously, if feasible. Combining electroporation modalities may also be an option to investigate a two-target approach.

## 6. Conclusions

ECT and IRE are becoming implemented in an increasing number of disease areas as larger clinical trials are being published. They have been shown to illicit ICD and have been linked to the abscopal effect. Moreover, preclinical studies investigating the use of ECT and IRE in combination with various immunotherapy regimes have shown promising efficacy in melanoma, pancreatic cancer, and hepatocellular carcinoma, which has led to the launch of several clinical studies. The most promising focus area is ECT + ICIs in patients with metastatic melanoma and IRE + ICIs or adoptive cell transfer using allogenic γδ T cells in patients with locally advanced pancreatic cancer and hepatocellular carcinoma. In the coming years, novel immunotherapy therapies such as IL-12 GET and TLR agonists, not only in combinations with electroporation modalities, but potentially also in triple regimens including ICIs should be investigated. Finally, meticulous patient selection for different combinatory regimens must be highly prioritized in order to optimize personalized treatment options.

## Figures and Tables

**Figure 1 cancers-14-02876-f001:**
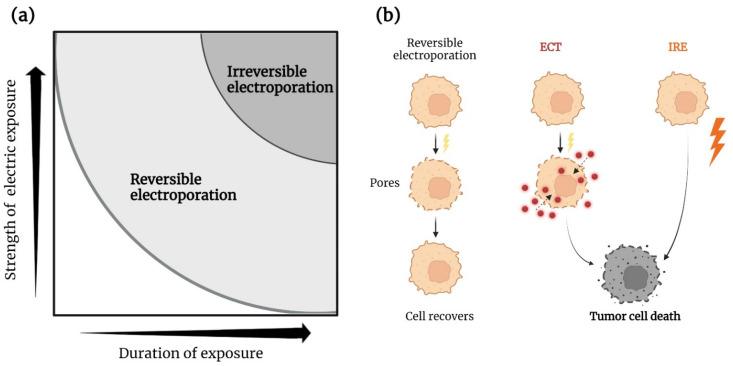
(**a**) The strength and duration of electrical stimulation determines the cellular outcome. (**b**) The addition of agents such as chemotherapy can prevent cancer cells from recovery and lead to cell death. ECT, electrochemotherapy; IRE, irreversible electroporation. Created with BioRender.com (accessed on 7 April 2022).

**Figure 2 cancers-14-02876-f002:**
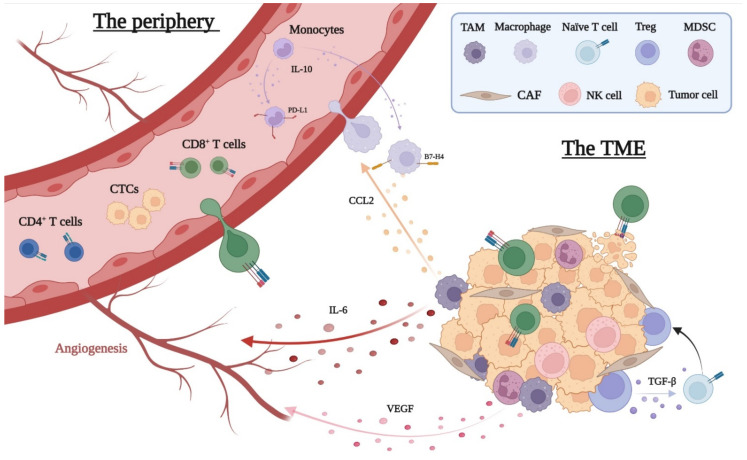
The interplay between the peripheral immune system and the tumor microenvironment (TME). CAF, cancer-associated fibroblast; CCL, CC chemokine ligand; CTC, circulating tumor cell; IL, interleukin; MDSC, myeloid-derived suppressor cell; PD-L1, programmed death-ligand 1; TAM, tumor-associated macrophage; Treg, regulatory T cell; VEGF, vascular endothelial growth factor. Created with BioRender.com (accessed on 7 April 2022).

**Figure 3 cancers-14-02876-f003:**
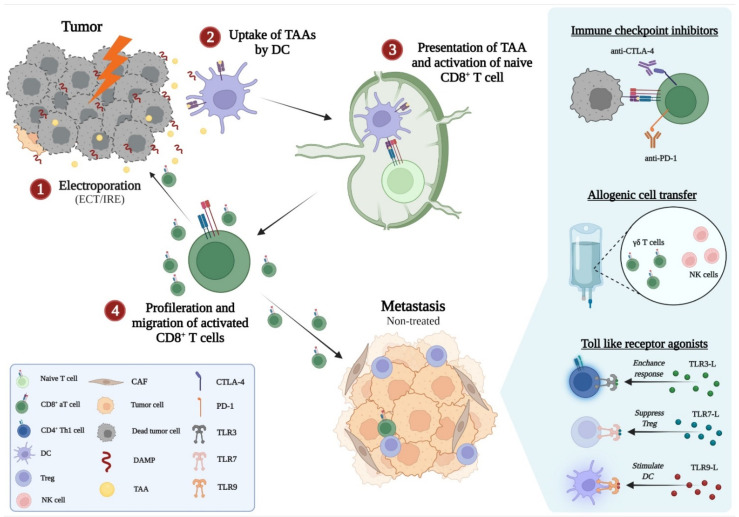
Model of how electroporation may induce an immune response and elicit an abscopal effect when combined with immunotherapy. aT cell, activated T cell; CAF, cancer-associated fibroblast; CTLA-4, cytotoxic T lymphocyte antigen 4; DAMP, damage-associated molecular pattern; DC, dendritic cell; ECT, electrochemotherapy; IRE, irreversible electroporation; NK cell, natural killer cell; PD-1, programmed death receptor 1; PD-L1, programmed death-ligand 1; TAA, tumor-associated antigen; TLR, Toll-like receptor; TLR3-L, TLR3 ligand; Treg, regulatory T cell. Created with BioRender.com (accessed on 7 April 2022).

**Table 1 cancers-14-02876-t001:** Summary of ECT studies investigating the effects on the immune system.

Species	Authors	Interventions (Type, n)	Cancer Types	Key Findings
Human	Gasljevic et al., 2017 [77]	ECT (bleomycin, 7)	Colorectal cancer	ECT induced coagulation necrosis. The majority of vessels >5 mm in diameter remained functional.
	Bigi et al., 2016 [74]	ECT (bleomycin, 2)	Cutaneous melanoma	High prevalence of tumor-infiltrating CD8^+^ T cells and foci of NK cells 3 h to 1 month after ECT. Apoptotic cell death was followed by necrosis 48–72 h after ECT.
	Gerlini et al., 2013 [79]	ECT (bleomycin, 9)	Metastatic melanoma	ECT promoted Langerhans cell migration from the tumor to draining lymph nodes and DC recruitment to the tumor. Further, DCs found in low number before ECT greatly increased at day 7 to 14.
Mouse	Tremble et al., 2019 [75]	ECT (cisplatin)	Colorectal cancer	ECT increased tumor infiltration of macrophages, neutrophils, B, NK, natural killer T cells, and DCs. Further, it decreased tumor growth of both treated and distal non-treated tumors.
	Ursic et al., 2018 [72]	ECT (cisplatin/oxaliplatin)	Melanoma	ECT induced a 4-fold increase in tumor infiltration of NK cells and CD8^+^ T cells.
	Calvet et al., 2014 [71]	ECT (bleomycin)	Colon cancer	ECT induced ICD through the liberation of ATP and HMGB1 and the translocation of calreticulin to the cell surface. Seven out of 8 immunocompetent mice were disease-free 24 days after ECT treatment, whereas all immunodeficient mice presented PD.
	Markelc et al., 2013 [80]	ECT (bleomycin)	Colorectal cancer	ECT induced a complete stop of the tumor blood vessels for up to 24 h. No damage to peritumoral normal blood vessels.
	Roux et al., 2008 [73]	ECT (bleomycin)	Sarcoma	ECT induced recruitment of tumor-infiltrating DCs and CD8^+^ T cells after 48–96 h, while the presence of CD4^+^ T cells remained stable.
	Torrero et al., 2006 [81]	ECT (bleomycin)	Breast cancer	ECT induced inhibition of angiogenesis in tumors but did not increase CD8^+^ T cell activity.
	Mekid et al., 2003 [82]	ECT (bleomycin)	Sarcoma	ECT increased the tumor infiltration of lymphocytes after 25, 50, and 75 h, in particular in the vicinity of apoptotic cells.
	Sersa et al., 1997 [70]	ECT (cisplatin)	Sarcoma	The tumor growth delay in immunocompetent mice was twice as long as in immunodeficient mice. Further, a high percentage of tumor cures was achieved in immunocompetent mice but none in immunodeficient mice. Of the mice cured after ECT, 75% rejected the tumor challenge, while none of the control mice did.
Cell	Fernandes et al., 2019 [83]	ECT (bleomycin/cisplatin/oxaliplatin)	Pancreatic cancer	ECT led to necroptosis.
	Ali et al., 2018 [84]	ECT (bleomycin/cisplatin/oxaliplatin)	Pancreatic cancer	The ECT treatments induced changes in stemness inducing factors related to cancer stem cells.

DC, dendritic cell; ECT, electrochemotherapy; ICD, immunogenic cell death; NK cells, natural killer cells; PD, progressive disease.

**Table 3 cancers-14-02876-t003:** Summary of ECT + immunotherapy studies.

Species	Authors	Interventions (n)	Cancer Types	Key Findings
Human	Campana et al., 2021 [124]	ECT (bleomycin)/pembrolizumab/ECT + pembrolizumab (41/44/45) **	Metastatic melanoma	Local response: ECT/pembrolizumab: 44%/32 CR, 37%/7% PR ECT + pembrolizumab: 49% CR, 29% PR Systemic response: Pembrolizumab: 21% CR, 4% PR ECT + pembrolizumab: 11% CR, 13% PR Two-year OS: Pembrolizumab: 43% ECT + pembrolizumab: 70%
	Quaresmini et al., 2021 [133]	ECT (bleomycin) + nivolumab (1) *	Metastatic melanoma	Durable CR (>1 year)
	Karaca et al., 2018 [134]	ECT (bleomycin) + nivolumab (1) *	Metastatic melanoma	Durable CR (>1 year) locally and systemic
	Hribernok et al., 2016 [126]	ECT (bleomycin/cisplatin) + INF-α (5) **	Advanced melanoma	Three patients with CR (1–23 lesions), 1 patient with CR of >85% of lesions (80 lesions), 1 patient with PR (5 lesions)
	Theurich et al., 2016 [123]	(ECT/radiotherapy) + ipilimumab/ipilimumab (45/82) ***	Advanced melanoma	Local response: Ipilimumab: 0% CR, 18% PR Ipilimumab + (ECT/radiotherapy): 7% CR, 31% PR Median OS: Ipilimumab: 42 weeks Ipilimumab + (ECT/radiotherapy): 93 weeks (hazard ratio 0.46)
	Heppt et al., 2016 [125]	ECT (bleomycin) + ICI (ipilimumab/pembrolizumab/nivolumab, 33) **	Metastatic melanoma	Local response: 15% CR, 52% PR Systemic response: 6% CR, 16% PR Median progression free survival: 2.5 months; median OS: not reached
	Mozzillo et al., 2015 [135]	ECT (bleomycin) + ipilimumab (15) **	Metastatic melanoma	Local response: 27% CR, 40% PR Systemic response: 0% CR, 33% PR One-year OS: 86% At week 10 and 12, a decrease in the absolute Treg number was seen in responders compared to no responders
	Brizio et al., 2015 [136]	ECT (bleomycin) + ipilimumab (1) *	Metastatic melanoma	ECT: Multiple liver and adrenal glands metastases after 3 ECT treatments ECT + ipilimumab: Durable CR (1 year) locally and systemically
	Andersen et al., 2003 [122]	ECT (bleomycin) + IL-2 (6) ***	Metastatic melanoma	ECT + IL-2 induced a partial remission Antitumor cytotoxic T lymphocyte responses declined following IL-2 therapy
Dog	Salvadori et al., 2017 [137]	ECT (cisplatin) + IL-12 GET	Mast cell tumor	Sixty-four percent CR Increased tumor infiltration of T lymphocytes at 4 weeks
Rabbit	Ramirez et al., 1998 [138]	ECT (bleomycin) + IL-2 secreting cells	Hepatocellular carcinoma	Median survival: Controls: 50 days (average number of metastases of 27) ECT: 82 days (average number of metastases of 18) ECT + IL-2: 80 days (average number of metastases of 3)
Mouse	Ursic et al., 2021 [127]	ECT (cisplatin/oxaliplatin/bleomycin) + IL-12 GET	Colorectal cancer Breast cancer Melanoma	ECT (cisplatin/oxaliplatin/bleomycin): 83%/83%/50% CR ECT + IL-12: 100%/100%/50% CR ECT: 50%/33%/17% CR ECT + IL-12: 67%/83%/33% CR ECT: 0%/0%/0% CR ECT + IL-12: 38%/0%/0% CR
	Tremble et al., 2018 [130]	ECT (cisplatin) + inducible T-cell co-stimulator	Colorectal cancer Metastatic Lewis Lung Carcinoma	Median survival: Control/ECT: 12/24 days ECT + inducible T-cell co-stimulator: 80 days ECT + inducible T-cell co-stimulator reduced the tumor growth of secondary non-treated tumors and increased the survival One hundred-day survival of 33% compared to 0% in monotherapy groups
	Cemazar et al., 2015 [131]	ECT (cisplatin) + TNF-α	Fibrosarcoma	Control/ECT: 0% CR ECT + TNF-α: 36% CR
	Sedlar et al., 2012 [128]	ECT (cisplatin) + IL-12 GET	Fibrosarcoma	Control/ECT: 0%/17% CR ECT + IL-12: 60% CR
	Roux et al., 2008 [73]	ECT (bleomycin) + CpG oligodeoxynucleotides	Fibrosarcoma Melanoma	ECT: Recruitment of tumor-infiltrating CD8^+^ cells 48–96 h after ECT; CD4^+^ cells remained stable Forty-three percent CR in treated tumors, 0% CR in non-treated tumors ECT + CpG: 100% CR in treated tumors, 57% CR in non-treated tumors ECT + CpG: Superior efficacy in reducing tumor volume compared to ECT, both in treated and non-treated tumors; induced a functional and specific activation of T cells both regionally (draining lymph node) and peripherally
	Torrero et al., 2006 [81]	ECT (bleomycin) + IL-12 GET	Breast cancer	Median survival: Control/ECT: 34/46 days ECT + IL-12: 60 days
	Kishida et al., 2003 [129]	ECT (bleomycin) + IL-12 GET	Melanoma	Median survival: Control/ECT: 18/37 days ECT + IL-12: 62 days In a metastatic model, ECT + IL-12 reduced the number of metastatic foci and increased the survival compared to monotherapy
	Sersa et al., 1997 [132]	ECT (bleomycin) + TNF-α	Fibrosarcoma	Median survival: Control/ECT: 24/33 days. 0% CR ECT + TNF-α: 50 days. 33% CR
	Mir et al., 1995 [139]	ECT (bleomycin) + IL-2 secreting cells	Fibrosarcoma	ECT: 60% CR ECT + IL-2: 100% CR Fifty percent CR in non-treated tumors; increased infiltration of CD4^+^ and CD8^+^ T cells in both treated and non-treated tumors

* case report; ** retrospective study; *** prospective study; CR, complete response; ECT, electrochemotherapy; GET, gene electrotransfer; ICI, immune checkpoint inhibitor; IL, interleukin; OS, overall survival; PR, partial response; TNF-α, tumor necrosis factor α.

## Data Availability

No datasets were generated or analyzed during the current study.

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
