# Peer review of "Electroporation and Immunotherapy—Unleashing the Abscopal Effect"

_cancers, 2022, doi:10.3390/cancers14122876_

Round 1

Reviewer 1 Report

“Electroporation and Immunotherapy – Unleashing the Abscopal Effect” is a review article written by Justesen et al. on the subject of electroporation and its use to induce an immune system response in cancer therapy. Overall, the review is very well written and informative. I find it almost ready for publication, with just a few minor comments.

-          Section 1, line 36 – the abbreviation for electrochemotherapy should be introduced here rather than at the end of the section.

-          A couple lines on the early use of electroporation prior to the development of electrochemotherapy in oncology would help flesh out the introduction a bit more (e.g. gene transfer). Also, electroporation is, to my understanding, just as prevalent (or more so) as a transfection tool – consider clarifying that cancer treatment is its major use within medicine.

-          Section 1.1, line 46 onwards – can you provide a brief summary on the significance of the abscopal effect on patient outcomes in those studies? Are survival rates significantly improved? In other words, how important is this effect to pursue?

-          Section 2, lines 69-78 – a reference for the mechanism of electroporation would be useful. Have you got a reference for the number/width/amplitude of pulses? Is it consistent across all relevant cell types, or do some applications/tumor types require different parameters? Can you elaborate/provide references for how these parameters were optimized? Also consider cleaning up the wording a little bit (‘V’ or ‘volts’ instead of ‘voltage’, 100 µsec width or duration).

-          Section 2, lines 89-90 (and also section 1.1, lines 48-51) – can you clarify/emphasise the relationship of the abscopal effect and immunogenic cell death? Is it mostly related to the release of DAMPs?

-          Related – the abscopal effect is not mentioned as a term anywhere between pages 11 and 21. Referencing it directly, or reinforcing the relationship of the abscopal effect and ICD in earlier sections would help improve the flow of the “story”. Figure 3, which illustrates it quite nicely, is currently not referenced anywhere in the text.

Table 2, Pandit et al. 2019 – the key findings section states “The peripheral Treg populations increased day 1 to 3 and decreased from day 3 to 5 in the IRE group compared to increase on day 1 to 3 and decrease on day 3 to 5 in the pancreatectomy group.” – this reads as if the effect of IRE and pancreatectomy on Treg populations was identical, can you confirm this is intended?

Reviewer 2 Report

Freyberg Justensen et al. wrote a comprehensive, detailed and understandible review on electroporation and immunotherapy and their role in in the remission of the tumors located out of the treated area. Paper is well organized, easily readible, equipped with a few informative tables and figures. The primary value and significance of the paper is a summary of all important findings in the field. References are appropriate.

There are only a few minor issues that have to be corrected:

1. page 7, lines 256- 257 - statement is not clear. One possible option is that there is a dot instead of comma and then the use of singular instead of plural. The second option is to erase that  short sentence in the lines  257-258 ("Cell which are all involved in the cancer immuno-surveillance.")

2. page 9, lines 313-314 - Statement "IRE treated tumors have shown increased CD30 expression levels" should be re-write. IRE treated tumors do not show increased CD30 expression levels per se although there is an increased infiltration of CD30+ lymphocytes after the treatment (see cited reference).

Reviewer 3 Report

1.       Please remove CaEP from the review as there is no preclinical evidence of the activation of the immune system after CaEP and one single case report in literature showing an abscopal effect after CaEP.

2.       Summary and abstract: ECT is commonly applied to treat cutaneous and subcutaneous tumours, not only melanoma, as shown by Clover AJP et al. in the European Journal of Cancer 2020 and by Plaschke et al. EJC. 2017, please modify the text accordingly.

3.       Paragraph 2.1: ECT has recently been proven to be valuable therapeutic option for liver and pancreatic tumours (Tarantino L et al. World J Gastroenterol, 2017; Edhemovic I et al.  J Surg Oncol, 2020; Izzo F. et al. Journal of Clinical Medicine 2021), please modify the text according to the literature.

4.       Paragraph 3.4: The authors describe the interaction between ECT and the immune system. The authors forgot to mention that the preservation of large blood vessels and the release of large quantities of tumour-associated antigens do apply also to ECT and not only to IRE (as described in paragraph 3.5), please modify the text according to the literature (Bigi L Clin. Cosmet. Investig. Dermatol. 2016; Gasljevic, G. PLoS ONE 2017; Zmuc, J. Sci. Rep. 2019).

5.       Page 7, lines 280-282: Papers from Roux S. et al. 2008 and Temble 2019 proove that ECT is able to induce distant tumor response, that is the definition of abscopal effect, so the literature is clear in proving that ECT can elicit an abscopal effect in animal models: please modify the text according to the literature.

6.       Page 14, line 410-411: please rewrite the sentence describing GET because it is not clear.
